# The practice of gender and protection mainstreaming in health response in humanitarian crisis - A case study from the refugee camps in Cox's Bazar, Bangladesh

Charls Erik Halder[1,2]*, Md Abeed Hasan[3]

1 Global Public Health Specialist, Chattogram, Bangladesh, 2 Humanitarian and Conflict Response Institute, University of Manchester, Manchester, United Kingdom, 3 Approaches to Interdisciplinary Research and Education, Université Paris Cité, Paris, France

* cehalder@outlook.com

## Abstract

### Background

The health system is required to be safe, equitable, and accessible to all ages, gender, and vulnerable groups, including older persons and persons with disability, and address their specific needs and concerns. However, limited evidence is available on the effectiveness and practicality of gender and protection mainstreaming interventions in health response in humanitarian crises.

### Objective

The overall objective of the research was to explore practices, gaps, and challenges and generate recommendations regarding gender and protection mainstreaming in health response to the Rohingya refugee crisis in Cox's Bazar, Bangladesh.

### Methodology

The research employed a qualitative case study design to explore the practice of gender and protection mainstreaming in health response in Cox's Bazar. Data collection methods include an extensive literature review and in-depth interviews with professionals. The professionals interviewed from the area of health and protection, specifically gender, child protection, emergency health intervention, and primary health activities. Data were analyzed using thematic analysis related to gender and protection mainstreaming. Limitations were assessed as to researcher bias because the researcher did all the coding; however, an open recording process, inter-literature cross-potentiation, and ethical considerations of research helped add to the reliability of the research. Exclusion criteria were defined to ensure data consistency, removing insufficiently detailed responses not pertinent to the research objectives.

**Data availability statement:** This is a qualitative study. All relevant information collected is generalized within the manuscript. All data are incorporated into the manuscript and/or the Supporting Information files. For any specific inquiries, the corresponding author may be contacted directly.

**Funding:** The author(s) received no specific funding for this work.

**Competing interests:** The authors have declared that no competing interests exist.

## Result

The study found a range of good practices on gender and protection mainstreaming in health response, e.g., placement of a gender action plan, monitoring system for gender and disability inclusion, emergency preparedness and response system, availability of sex-segregated toilets and waiting spaces, availability of gender-based violence service and engagement of female community health workers. The study also revealed some best practices which have the potential to scale up, e,g. psychosocial spaces at health facilities for children, palliative care for terminally ill patients, integrated medical and protection services, and facilitation of community health facility support groups. Critical gaps were found in the areas of women's leadership, coordination, capacity building, targeted interventions for vulnerable groups, infrastructural adaptation and consultation with the community on their concerns.

## Conclusion

We urge policymakers, sector coordinators, health program management, healthcare workers, and global stakeholders to address the gaps and challenges, learn and scale up the best practices, and take action to implement the study's recommendations to maximise gender and protection mainstreaming in health response.

## Background

In humanitarian crises, health response aims to reduce avoidable mortality, morbidity and disability that could result from the crisis and to restore access to preventive and curative healthcare [1]. The humanitarian principle requires that any humanitarian aid, including health assistance, should be provided based on need and without discrimination [2], meaning that everyone should have meaningful access, safety and dignity in humanitarian health service [3]. Therefore, the health system requires it to be safe, equitable and accessible to all ages, gender and vulnerable groups, including children, older persons, men, women, gender-diverse populations (GDP) and persons with disability, and address their specific needs and concerns. In a word, gender and protection should be mainstreamed with health response in humanitarian settings, which means that the health response should prioritise safety and dignity, avoid causing harm, ensure meaningful access of all people to health service without any discrimination, promote accountability to the affected population and assist people in claiming their rights to health [3]. Gender and Protection Mainstreaming is the process of integrating gender considerations and protection principles across all aspects of humanitarian programming to ensure equitable access, participation, and safety for all individuals, particularly vulnerable populations such as women, children, older persons, and persons with disabilities. This approach ensures that assistance does not cause harm, promotes dignity, and addresses specific risks and needs within crisis-affected communities [4,5].

Humanitarian crises disproportionately affect women, girls and children and put them at a heightened risk of gender-based violence (GBV) due to breaches of security and protection measures [6]. In some crises, more than 70% of women have faced GBV [6]. Patriarchy and discriminatory social norms increase the vulnerability of women to disease, disability and injuries and impede their access to healthcare [7,8]. Globally, approximately 15% of the population has some form of disability, and around 13% of the people are aged over 60 [9]. Persons with disability and older persons often face discrimination based on their disability and age. In many cases, the inclusion of persons with disability and older people remains under-prioritised, and health partners may lack the capacity to improve their access and respond to their unique needs [9,10]. People with other vulnerabilities, e.g., people living with HIV (PLHIV) or sexually transmitted infections (STI), may be stigmatised in society, affecting their dignified access to health services.

This qualitative case study was implemented in the context of the Rohingya refugee crisis in Cox's Bazar, Bangladesh, one of the fastest-growing and largest refugee crises in the world. An estimated 925,380 Rohingya refugees are residing in 33 extremely congested camps in Cox's Bazar following their mass displacement from Myanmar in 2017, ignited by systematic discrimination and targeted violence [11]. The refugee crisis resulted in an increased demand for health services, which is being further exacerbated by frequent outbreaks of infectious diseases, like diphtheria, measles, cholera, COVID-19, and dengue [12–14]. To meet the current demand, 80 health partners are providing primary and secondary healthcare services in the camps through 93 health posts, 43 Primary Health Care centres (PHCs), Field Hospitals and a network of 1300 community healthcare workers (CHW) [15]. The gender and protection concerns mentioned above also mirror in this crisis. Rohingya women and girls have the added risk and challenges of insecurity, violence, restricted mobility or speaking up and lack of influence on decision-making [16]. An estimated 12% of the population has some form of disability and faces multiple barriers, discriminations and limitations to access care and services, and older female persons with disabilities were disproportionately affected by the gaps and challenges [17].

To ensure the mainstreaming of gender and protection in health response in humanitarian crises, many guidelines and tools are made available for the humanitarian partners both globally and in specific local contexts, especially in Cox's Bazar. Global protection cluster [18], as well as protection clusters of different humanitarian crisis settings, including Pakistan [19] and Somalia [20] as well as UNICEF [21] produced guidelines on protection mainstreaming in humanitarian health response. The guidelines highlighted a wide range of measures, including safety and dignity, gender-and disability-inclusive infrastructure, accessibility of health services to all, including persons with disability and the elderly, prevention of and response to gender-based violence, staffing representatives of gender, ethnic and economic differences, provision of reproductive and obstetric health services, provision of private breastfeeding spaces, women's empowerment and leadership, provision of community outreach activities for women and girls, and interventions for culturally appropriate psychosocial support. Considering the inclusion of persons with disability and older people is not adequately prioritised in humanitarian health response, IASC (Inter-Agency Standing Committee) guidelines on the inclusion of persons with disability in humanitarian action [22] and the Humanitarian inclusion standards for older people and persons with disability [9] clearly outlined the actions and standards for the inclusion of older people and persons with disability, specific to health response. Locally, in the context of the Rohingya refugee crisis in Cox's Bazar, several guidelines are also available for gender and protection mainstreaming in humanitarian health response.

However, as per the author's literature review, no study was found that comprehensively examined the practical aspects of overall protection and gender mainstreaming in humanitarian health response except some isolated studies on women's leadership, sexual and reproductive health services, prevention of and response to sexual and gender-based violence (SGBV) and inclusion of persons with disability and older persons. For instance, although enhancement of gender-inclusive leadership and engagement of women frontline workers, women groups and networks in decision-making can contribute to achieving better health outcomes for the affected population in humanitarian crises, women are found extremely underrepresented in humanitarian leadership [23,24]. In some fragile and post-conflict contexts, women are primarily employed in nursing and midwifery positions; however, they are grossly under-represented in the

management [25]. Female staff in humanitarian contexts often face challenges in balancing their career and cultural expectations of household activities [26].

Strong leadership and effective resource mobilisation, engaging with communities through outreach initiatives and tiered community reproductive health services, deployment of female community health workers, psychosocial interventions to address emotional and mental health needs and capacity building of healthcare workers, including those at lower levels are among the best practices in sexual and reproductive health (SRH) interventions evidenced in different humanitarian settings [27–32]. Some barriers in this field identified were lack of training, weak communication, inadequate staffing, limited attention to local organisation and insufficient focus on clinical management of rape services [27].

In summary, both globally and in the local context of Cox's Bazar, adequate guidelines and tools are available for the humanitarian partners for protection and gender-inclusive health interventions in humanitarian response. However, except in the area of SRH services, minimal studies have been made available that have generated evidence on the uptake, practicality, and effectiveness of those interventions. The research gap is more prominent for the Rohingya refugee crisis in Cox's Bazar. Despite the availability of general guidance on the inclusion of persons with disability and older people in humanitarian response, there are significant research gaps in evidence regarding the effectiveness of inclusion efforts, use of disability and age-disaggregated data over 60 years, cost and benefits of inclusion strategies, and the intersection of gender, disability and older age [9,33]. This study was undertaken to fill up the research gap with the overall objective of exploring practices, gaps, and challenges and generating recommendations concerning gender and protection mainstreaming in humanitarian health response to Cox's Bazar Rohingya refugee crises.

## Research methodology

### Study design

This was a case study with a qualitative design aimed to explore the practices, gaps, and challenges and generate recommendations concerning gender and protection mainstreaming in humanitarian health response. Given that the protection and gender mainstreaming is a complex phenomenon which involves awareness, actions, behaviour and capacity of a broad spectrum of stakeholders, including policymakers, humanitarian workers and the community, a qualitative approach was chosen to explore ongoing interventions. This approach helps assess how practical or effective these interventions are, what is working well, what challenges remain, and how the system can be further improved. Further, a qualitative case study is well suited to get extensive and in-depth explanations of a complex social phenomenon [34]. Given the lack of studies on the practicality of gender and protection mainstreaming in humanitarian health response, we conducted a case study in Cox's Bazar to gain a comprehensive understanding of this complex issue.

### Methods and approach

This qualitative study design is underpinned by a critical realism epistemological position. The research investigated the experiences of health and protection sector experts to understand the complexity of gender and protection mainstreaming practice in humanitarian health response, which was triangulated with existing literature/reports.

### Research setting

The study was implemented in Cox's Bazar district of Bangladesh. This district is inhabited by around 1 million Rohingya refugees living in 33 congested camps. This is one of the most significant refugee crises in the world and has a cluster-like humanitarian response involving multiple sectors and working groups, including health, protection, Water, Sanitation, and Hygiene (WASH), site management and communication with the community (CWC) [35]. More than 60 humanitarian organisations, including a mix of local, national and international organisations, are responding to this crisis under the umbrella of Intersector Coordination Group (ISCG) [36]. Thus, a case study on gender and protection mainstreaming in

this environment can represent comparable humanitarian situations worldwide, and the results can help enhance gender and protection mainstreaming in health in current and future crises globally.

### Data collection methods and processes

Two qualitative methods were deployed for this study – a) secondary data collection through an extensive literature review and b) primary data collection through in-depth interviews of 12 key informants. Data analysis was performed using thematic coding, with categories derived inductively from the collected data. The analysis was conducted solely by the lead researcher, which introduces the possibility of bias. To mitigate this, multiple iterations of coding were performed, and triangulation with existing literature was applied. To further mitigate subjective bias, a structured coding framework was followed, and preliminary results were shared with a senior researcher for validation before finalizing themes. The data collection/recruitment period was between 15 July 2022–15 August 2022.

a) **Literature review:** In addition to a generic literature review during the design phase of this study, an extensive literature review from the local context was done as part of data collection. This included a review of existing policies, strategies, guidelines and reports that focus or reflect on gender and protection mainstreaming in health response. The literature review aimed: a) to provide analysis of existing policy/strategic recommendations for gender/protection mainstreaming, b) to provide themes of discussions for in-depth interviews, and c) to enable triangulation and comparison of the findings of in-depth interviews in contrast to the existing recommendations/reports

b) **In-depth interviews with key informants**: Key informant interviews were conducted with experts from the health and protection sectors. *Sampling*: Purposive and snowball sampling techniques were employed to select the participants who have specialisation and expertise in the areas of a) gender, child protection and general protection; b) emergency health response; c) operation and management of primary health care; and experience with the Rohingya refugee crisis. The total sample size for the qualitative interviews was 12. Eight participants were initially selected purposively from the health and protection sector partners, and the remaining four were selected through the snowball technique. A sample size of 12 participants was chosen based on theoretical saturation, commonly used in qualitative research to ensure that the collected data encompasses all relevant themes. Since qualitative research prioritizes depth of insights over generalizability, statistical power calculations are not typically required. The adequacy of the sample size was assessed based on data saturation, ensuring that additional interviews did not yield new themes [37]. This approach aligns with best practices in qualitative research, where thematic saturation is a key determinant of sample sufficiency rather than statistical power. According to Guest et al. [2006], saturation is typically reached after conducting 12 interviews in homogeneous populations, and our findings align with this standard.

**Data collection instrument.** An interviewer-administered semi-structured questionnaire was used for data collection from the participants. The question had open-ended questions focusing on the experience of the participants in gender and protection mainstreaming in health response, particularly a) existing practices, b) practicability, effectiveness and efficiency of the practices, c) available guidelines and tools, d) competency, relevance and practicality of the tools, e) gaps and challenges, f) best practices, and g) recommendations for further improvement. The tool was piloted among 2 participants. During interviews, participants were directed by the themes highlighted in the literature review while leaving questions open-ended and retaining their natural social world.

**Data collection process.** One-to-one in-person or online interviews were conducted. Using the semi-structured questionnaire (S1 File Interview Guidance), each participant was asked about their experience, insight and opinions on existing and proposed practices of gender and protection mainstreaming in health response. The interviews lasted for about 1–1.5 hours and were arranged in a preferred private setting of the participant (e.g., office room of the participant, residence) in Cox's Bazar city or using Microsoft Teams. Interviews were conducted in English and Bangla

as per the convenience of the participant. In in-person interviews, data were scripted into a Microsoft Word document. In the case of online interviews, the recording was done by the Microsoft teams, and after transcription, the recordings were permanently deleted. To mitigate potential biases during data collection, we ensured that all interviews followed a structured questionnaire. Interviews were conducted in a private setting to encourage honest responses, and participants were informed of their right to withdraw at any time. For interviews conducted in Bangla, professional translators ensured the accuracy of transcriptions. To maintain confidentiality, all transcriptions were stored in a secure system, and personal identifiers were removed.

## Data analysis

Recommendations from the existing policies and strategies were thematically organised. As mentioned, interviews were transcribed verbatim in a Microsoft Word document. We employed thematic analysis for qualitative data collected through in-depth interviews. The in-depth interviewed were analysed using thematic analysis, as the methods is appropriate for exploring experiences, thoughts and behaviours [38]. As this study is qualitative in nature, statistical assumptions regarding univariate and multivariate normal distribution do not apply. Instead, the thematic analysis approach was selected as it allows for in-depth exploration of participant perspectives without reliance on parametric statistical assumptions [39]. As guided by Braun and Clarke [40], data were manually analysed in six steps: a) familiarisation with all the data through reading and re-reading, b) coding key features into the entire data set, c) collating the codes into potential themes; d) reviewing the themes against the codes and dataset and generating a thematic map; e) defining and naming the themes, and f) writing the narrative of analysis. During the analysis process, themes were generated inductively from the interview data as well as deduced from the literature review. Validation of the data was an integral part of the data analysis process. Items with low relevance to the research questions/themes were eliminated, and items having similar concepts were merged.

## Ethical consideration and data confidentiality

The research project was reviewed and approved by the School of Arts, Languages and Cultures Ethics Committee at the University of Manchester (Ref: 2022-13668-24736). The study was designed in such a way that it eliminated any risk of physical or mental discomfort, harm, or damage from its process or publication and placed the highest priority on protecting the rights and welfare of research participants as well as vulnerable refugees in the target settings. The research is planned based on morality to address the evidence gaps on gender and protection mainstreaming in humanitarian health response and generate recommendations for improvement in this field in local and global humanitarian contexts. No commercial or financial relationships could be taken as a potential conflict of interest.

   The study did not collect any personally identifiable information (except for the consent form) or any sensitive or confidential information. The study did not involve participants from vulnerable or dependent groups, such as Rohingya refugees, considering their protection concerns. Before the interview with the professionals, the participants were provided with clear information on the background, objectives, and study procedure. Participants were given an information sheet detailing the purpose of the study, potential risks and benefits, voluntary participation, and the right to withdraw at any time without providing a reason. Informed consent was obtained through a signed form before the interview commenced. In cases where participants experienced emotional distress during or after the interview, they were informed of the availability of psychological support. Participants were provided with a list of mental health and psychosocial support (MHPSS) resources, including contact details of professional counselors affiliated with humanitarian organizations operating in the region. This information was explicitly mentioned in the consent form and information sheet to ensure participants were aware of available support services. The study followed the Inter-Agency Standing Committee (IASC) Guidelines on Mental Health and Psychosocial Support in Emergency Settings [22], which emphasize the ethical responsibility of researchers to address potential distress and provide necessary referrals for support. To ensure confidentiality,

no identifying personal information was recorded in the transcripts, and participants were assigned numerical identifiers instead of names. Anonymity was maintained by removing any contextual references that could reveal the participant's identity. All electronic files, including interview transcripts and consent forms, were securely stored in an encrypted folder with restricted access, and no data was shared beyond the research team. Written informed consent was taken from each participant. Participants were allowed to skip any question or withdraw themselves from the interview at any time. All data were stored in the university-provided Pen-drive, and data analysis was done on an encrypted computer. In the case of online interviews, the recordings were permanently deleted from the storage once transcription writing was completed. Identifying information was stored securely and separately from the study data. The information was kept in a password-protected file.

## Results

This research explored current strategies, guidelines and practices of gender and protection mainstreaming and the practicality of different interventions in healthcare settings, including the best practices, gaps and challenges. The participants were categorized into three groups: (1) healthcare professionals (e.g., medical doctors, and health program managers), (2) protection sector specialists (e.g., gender-based violence coordinators and child protection specialists), and (3) humanitarian policymakers involved in health coordination. Their experience ranged from 5 to 20 years, with 50% representing UN agencies, 30% from international NGOs, and 20% from local NGOs. These details ensure transparency regarding the selection criteria and expertise of participants, strengthening the validity of findings. In this section, firstly, the findings from the review of local policies, strategies and guidelines are presented, and the gender and protection mainstreaming interventions recommended by relevant strategies and guidelines are summarised into a thematic framework. To ensure a structured flow, we first summarize key findings from the literature before transitioning to primary data from in-depth interviews.

This approach contextualizes participant perspectives within the existing research and enhances coherence.

### Policy, strategies and guidelines

A significant number of strategies, guidelines and resources are available in Cox's Bazar to guide the humanitarian partners on gender and protection mainstreaming in health response. Gender and protection mainstreaming is strategised in the health sector strategic plan through its commitment to prevent and respond to GBV, strengthen mainstreaming of accountability to affected population (AAP) in all phases of health response, promote accessibility and inclusion of persons with disability and prevent sexual exploitation and abuse (PSEA) [15].

The health sector gender action plan (GAP) outlined targeted and gender mainstreaming interventions in line with the overall objectives of the sector [41]. The plan focused on generating sex, age, and disability disaggregated data and reports, monitoring the provision of GBV and gender-responsive health services, establishing gender-responsive health facilities (e.g., sex-segregated rooms, female staff), engaging women and community leaders on gender-responsive health operation, capacity building of staff and volunteers on gender, GBV, protection and PSEA and updating the partners on needs, challenges and responses concerning gender-responsive health service. The plan provides specific indicators in these areas to monitor the interventions [41].

The health and protection sectors jointly formulated a tip sheet for protection mainstreaming in health response for the Rohingya refugee response in line with the Global Protection Mainstreaming Health Tip Sheet [42]. To prioritise safety, dignity and avoiding harm, the document guided safe location and routes of health facilities, infrastructural adaptation, prioritisation of vulnerable groups for care, general awareness raising on PSEA, respectful and inclusive cultural and religious practices, confidentiality and privacy, and deployment of female staff. It provided tips to ensure meaningful access to health services by all, inclusive of persons with disability, older persons and mental health conditions, suggested having staff representative of gender and ethnic differences, and outlined action points for health

 

staff to respond to specific needs of victims of human rights violence including GBV. The guideline also outlined actions to ensure accountability and participation of vulnerable individuals through coordination with local authorities and civil society organisations working with older persons and persons with disability, facilitation of health committees with representation from all layers of society, capacity building of staff on communication with children and establishment of suggestions and complaint mechanism [42].

The health sector developed a framework for ensuring AAP in line with Global Health Cluster Operational Guidance on AAP to mainstream accountability of health partners to patients, beneficiaries and their communities [43]. The framework focused on the engagement of all layers of people in all phases of the health program, enhancing awareness of patient rights, communication with the community and establishment of feedback and response mechanism [43].

The Health Sector and Child Protection subsector jointly developed guidance for child protection and health care for children in health facilities during the COVID-19 outbreak, addressing their vulnerabilities [44]. This is detailed under the child protection section later in this article.

Application of Gender with Ager Marker (GAM) [45] is a prerequisite for submitting a project proposal by the health partners to the Joint Response Plan (JRP), i.e., the humanitarian appeal, to assess how their projects contribute to enhancing opportunities for different gender, age, and disability groups through meaningful and effective engagement. The health sector partners are trained by the Gender in Humanitarian Action (GiHA) Working Group to effectively apply the tool in their applications.

The framework outlines the gender and protection mainstreaming in health response activities recommended by the local policies, strategies, and guidelines in Cox's Bazar (Table 1).

## Coordination and partnership

The health sector implements several initiatives for gender mainstreaming in line with its gender action plan, which includes sharing the Gender Action Plan with the partners, collecting and disseminating gender-disaggregated reports and supportive supervision of health facilities for their adherence to recommended gender-responsive measures. Linked to the Gender Action Plan and other social inclusion actions, the health sector also performs quarterly monitoring of the actions, e.g., the presence of sex-segregated toilets, measures for access for persons with disability, availability of GBV services, presence of community complaints and feedback mechanisms. Although this monitoring system contributes a lot to the partners' adherence to the recommended actions, some of the indicators in the system only focus on physical presence/availability rather than quality and/or functionality. As expressed by one participant, *"…for instance, the monitoring system finds out whether a health facility has a disability-friendly toilet, but the system cannot track whether the toilet is actually being utilised by a person with a disability".*

For the Rohingya refugee crisis, there is a Gender in Humanitarian Action Working Group (GiHA WG) that provides support to all sectors and humanitarian organisations for effective integration of gender in their actions through coordination, technical advice, guidance, capacity building, advocacy, assessment, and information sharing [46]. The health sector collaborates with GiHA WG for the strategic review of gender mainstreaming in health response and capacity building of the partners in the aspect of gender integration (e.g., Gender with Age Marker) and commemoration of relevant days (e.g., International Women's Day 2022).

There is also close collaboration between the health sector (and its relevant working groups, e.g., SRH working group) and the GBV subsector of the protection sector for establishing and implementing the GBV referral pathways at the refugee camps, including capacity building of the partners on compassionate GBV referrals. However, as per some participants, limited partnership and networking efforts are observed in terms of protection of other vulnerable individuals, especially persons with disability and older people. Only a few agencies, e.g., Humanity & Inclusion and Help Age International, in Cox's Bazar, have dedicated health programs for persons with disability and older people.

**Table 1. Actions for gender and protection mainstreaming in Cox's Bazar recommended by policies and strategies**.

| Themes | Recommended interventions |
| --- | --- |
| **Coordination and partnership** | Develop, implement, and monitor the health sector gender action plan; discuss gaps/challenges in health sector coordination meetings. |
| | Develop minimum standards to guide partners on disability inclusion in healthcare. |
| | Coordinate with local authorities and Civil Society Organizations specialised in working with persons with disability to include them in the assistance program. |
| | Coordinate with the Age and Disabilities Working Group (ADWG) to map out NGOs offering services to vulnerable groups for training and referral of health staff. |
| | Distribute advocacy/joint communication products on gender-responsive programming and GBV. |
| | Update Health Sector partners and key health networks on gender-responsive health developments. |
| | Sensitize JRP partners for adherence to PSEA principles |
| | Collaborate with the protection sector, GBV and child protection subsectors, ADWG and Gender in Humanitarian Action working group |
| **Staffing, leadership and capacity building** | Staff should be representative of gender and ethnic differences; employ an adequate number of female health staff members; |
| | Staff capacity building on gender, child and protection mainstreaming, including linkage to psychosocial support and legal support |
| | Staff should speak in Rohingya language or be aware of common local vocabulary appropriate for the work. |
| **Safety and security:** | Safe and accessible location of health facilities and routes |
| | Consider seasonal environmental risks for accessing health facilities - consider transport arrangements. |
| | Place security measures for health facilities and routes (e.g., lights, security) |
| | Safe placement of medical supplies to avoid harm to children |
| | Placement of security guards, if necessary, for safety and protection |
| | Safe disposal of healthcare wastes |
| | Place crowd control and infection prevention and control measures |
| **Infrastructural adaptation, WASH and patient flow** | Separate waiting areas for male and female |
| | Infrastructural adaptation to health facilities and latrines for older persons and persons with disabilities (e.g., smooth paths, ramps) |
| | Prioritisation of older persons with disability, unaccompanied children and pregnant women in the queue |
| | Dignified waiting space for people with specific vulnerabilities (e.g., persons with disability, pregnant women, breastfeeding mothers, children, PLHIV and gender-diverse populations (GDP)) |
| | Separate male and female toilets in the health facility with appropriate labelling |
| | Child-friendly latrine design |
| | Separate shade for children and lactating mothers; breastfeeding corner |
| | Menstrual hygiene management kit available at health facility with training to staff |
| | Separate examination room with proper privacy |
| | Sex-segregated spaces and rooms at health facilities |
| **Child protection** | Signing of code of conduct by staff for child safeguarding |
| | Child-friendly waiting space; Space for children to play if they wait for caregivers. |
| | Child-friendly communication by health staff, including guards |
| | Ensure access to health services by children. |
| | Train staff on child protection, communication with children, and referral mechanisms for child protection cases. |
| | Explanation to children/caregivers before physical contact with children |
| | Contact with child protection actors for engagement with children without a caregiver. |

*(Continued)*

**Table 1.** (Continued)

| Themes | Recommended interventions |
|---|---|
| **Targeted interventions** | Special arrangements for individuals who cannot access health services, e.g., mobile health teams or home-based care. |
| | Establish a referral system for victims of human rights violations to specialised services, including MHPSS. |
| | Special service provision for consultation/treatment for persons with hearing, speech, language, psychosocial and/or intellectual disabilities. |
| **Prevention of and response to GBV (including PSEA)** | General awareness raising on (PSEA) rules and code of conduct |
| | Train staff on GBV/harmful traditional practices, response and referral pathways and psychosocial support |
| | Health facilities to collaborate with GBV and general protection partners for referral of GBV survivors |
| | Facilitate meetings among health, protection, GBV, and child protection actors on the referral pathways. |
| | Consider room design, type of furnishings and equipment for examination/consultation of GBV survivors. |
| | Place guidelines and mechanisms for monitoring and reporting abuse or exploitation |
| | Train staff on PSEA, including security guards and volunteers |
| | Present PSEA messages in a child-friendly and gender-sensitive manner |
| | Aware staff of do-not harm by recalling memories during psychosocial treatment |
| | Safety audit to identify current and newer risk |
| | Survivor-centred approach for all patients |
| | Report and share protection concerns of individuals to the protection sector, including GBV and child protection subsector |
| **Communication with the community and accountability to the affected population** | Design and implement health activities with community engagement in all phases of the project lifecycle |
| | Functional context appropriate (safe, accessible, trusted) feedback and complaint mechanism in an accessible place with multiple options and proper privacy |
| | Feedback mechanism flexible for children, illiterate people, persons with disability and older persons |
| | Consultation and meaningful engagement with all layers of the community (e.g., women, men, girls, boys, children, elderly, GDP and persons with disability) to identify and respond to the health needs |
| | Ensure consistent participation of all layers of the community and integrate their feedback during assessment, planning, implementation, monitoring, and evaluation. |
| | Ensure health committees and workers are representatives of all layers |
| | Training health committees on child protection, GBV, disability and diversity |
| | Engage women, community and religious leaders in gender-responsive health operations. |
| | Wide range of IEC materials (e.g., pictorials, audio, video, infographics) etc. to reach out to persons with non-mobility-related disabilities |
| | Support cultural, religious and social activities and practices that can impact safeguarding health. |
| | Ensure that information on waiting times, processes, and procedures at a facility is shared in the local language. |
| | Let beneficiaries know their rights to healthcare and how to access it. |
| | Consider local solutions/coping strategies to address the vulnerabilities (e.g., pooled funds for taxi services) |

*(Continued)*

**Table 1.** (Continued)

| Themes | Recommended interventions |
|---|---|
| Information management | Set project indicators and reports disaggregated by age, sex, gender, location and specific groups (e.g., persons with disability and ethnic minorities) |
| | Monitor and review the proportionate distribution of health services. |
| | Confidential handling of information – collection, processing, storing or communicating |
| | Confidentiality and privacy during consultation, counselling or personal information sharing |
| | Informed consent for sharing any information |

*Synthesized from a) Health Sector Strategic Plan (2023–2024) [13]; b) Health Sector Gender Action Plan [41]; c) Health Sector Programme – Tips for Protection Mainstreaming [42]; d) Accountability to Affected Populations (AAP) Framework [43]; and Child Protection & Health Care for Children in Health Facilities during COVID-19 Cox's Bazar [44]

There is also a gap noted on the unavailability of a contextualised framework for the inclusion of persons with disability and older persons, as expressed by an emergency health expert –

*"…there are some guidelines available globally for age and disability inclusion. For this context, it is required to have guidance for the partners on what minimum practical measures should be placed at the health facility as well as other health programs, for example, immunisation for improving access of persons with disability and older people".*

### Staffing and leadership

The guidelines recommend employing female health staff with skills and experience in working with women and children and ensuring that the health facilities have both male and female doctors and nurses [42]. According to the Health Sector Quarterly Monitoring Report (2022), facility monitoring and supervision visits confirmed that all health facilities (100%, n = 136) maintain at least one female medical professional during operational hours [47]. Participants highlighted that the presence of female staff in health facilities enhances women's comfort, including speaking up during consultations and participating in physical examinations.. Our findings reflect participants' perceptions and experiences, which were cross-referenced with existing guidelines and reports to ensure validity. Several participants noted that women utilize healthcare services more frequently than men in the Cox's Bazar refugee context, as reflected in facility-level service utilization reports.

In Bangladesh, nurse and midwife positions are traditionally seen as occupations for women. Therefore, naturally, most nurses and all midwives in the health facilities in the camps are women. However, females are still grossly lagging in other health professions, e.g., doctors, medical assistants, and laboratory technologists. Female representation remains scarce in leadership positions, such as facility managers or coordinators. At the time of the study, it was found that only two of the twelve camp health focal persons and field coordinators who coordinate camp-based health sector responses were female. While women have more significant leadership in the SRH Working Group, they have less representation in other working groups and technical committees. During the study, most of the working groups, e.g., Strategic Advisory Group, MHPSS working group, Emergency Preparedness and Response Technical Committee, Mobile Medical Team Working Group, and Community Health Working Group, were led by men. It can be related to the lack of family and social acceptance for women to work in humanitarian crises and the fact that there is no particular prioritisation system in place for women to be in leadership roles. As expressed by a participant, *"Although it is written in the job advertisements that women are encouraged to apply, often there is no mechanism in place for prioritising us (women) in different key roles."*

One gender expert expressed that *"…a reason (for lack of women leadership) can be their (women's) families and societies do not allow them or feel comfortable for them to work in a crisis setting, like Cox's Bazar".*

It is recommended that staff should be trained on different aspects of gender and protection mainstreaming, including gender diversity, gender-based violence, CMR (Clinical Management of Rape), child protection, disability and diversity, and PSEA mainstreaming [42]. Many of these trainings, e.g., PSEA and gender mainstreaming, are mandatory for all staff working in the UN agencies or international organisations. As a public health expert expressed,

> *"gender mainstreaming, PSEA, and code of conduct training are either mandatory or routinely organised for UN agencies and international organisations. However, this practice should be standardised for all humanitarian agencies, especially for local NGOs. The training methods should be adapted based on the different levels of capacity and skill of the staff or healthcare workers. For instance, a webinar could be helpful for office staff, but more practical and simplified training is required for ambulance drivers and cleaners, who are often third-party contracted".*

Participants noted that only a few training initiatives have been implemented for healthcare workers or health program managers/officers on age and disability inclusion, child protection, and protection of other vulnerable groups (e.g., gender-diverse populations, patients with HIV or Hepatitis C).

### Safety and security

While most facilities are located in an accessible location, due to the area's hilly terrain, there are a few facilities located on hilltops or at a location inaccessible by ambulances. This affects the accessibility of vulnerable individuals, especially pregnant women, persons with disability, older persons, and critically ill patients. The health sector and its partners conducted a rationalisation exercise to identify such facilities and recommend them for relocation to a safer and easily accessible location [48]. Some partners, after receiving recommendations from either the community or the sector, relocated their facility to a safer and more accessible location.

Apart from this, heavy monsoons resulting in flash floods sometimes impede the accessibility of the general community, especially vulnerable individuals, to health facilities. To mitigate the risk, the health sector established a mechanism for early notification of the closure of any facility due to monsoon or weather-related events. In such cases, a Mobile Medical Team can be activated to provide care to the affected community from an alternate site. However, in reality, the floods affect not only the accessibility of the community but also the healthcare workers travelling to the camps. During monsoon season, if the flood impedes the accessibility of healthcare workers to the camp, the health facility may be closed, resulting in the unavailability of health services for the affected population.

As recommended in the protection mainstreaming guideline [42], most facilities have placed security measures, e.g., lights, security guards, etc., to avoid any incidence of violence and to promote safer access to the facility by the beneficiaries, especially those with vulnerabilities. One of the challenges expressed by some participants is that the power supply is not always consistent in the camp setting. Most facilities rely on solar systems or electric generators. Rainy weather and mechanical disturbances of the solar system/generator may affect the solar/generator function.

Usually, medical equipment and supplies are kept in a safe location away from children to avoid any possible harm to them. However, participants expressed that the pharmacist and logistic staff should be further sensitised on this issue.

Further, participants stated that a good infection prevention and control (IPC) system has been placed in most health facilities in Cox's Bazar. World Health Organization provides training and technical assistance to the facilities for implementing IPC measures to prevent hospital-acquired infection among patients, healthcare workers, and the community. Healthcare waste management is still a critical concern for health facilities. However, partners are now more sensitised and trying to adopt environmentally friendly measures for biomedical waste management to prevent any harm or exposure to the patients and the community.

**Infrastructural adaptation, WASH, and patient flow**

In line with the protection and gender mainstreaming guidelines [42], participants opined that most health facilities have physically separated and labelled toilets for males and females. This is also evident in the health sector quarterly monitoring report suggesting that 90% of the health facilities in the camps had sex-segregated latrines for patients [47]. However, 30% of the facilities did not have sex-segregated latrines for staff [47]. Participants also suggested that some agencies have the practice of placing a menstrual hygiene kit in the female toilets for patients and staff. However, this practice is not widely used by the partners and is not adequately monitored.

Usually, most health facilities have segregation in waiting spaces for males and females, but due to high patient turnover and space limitations, it is often difficult to have separate consultation rooms for males and females, especially in small health posts. Participants opined that such a separate arrangement is needed in this setting to increase women's access to healthcare services. Prioritisation of older people, persons with disability, unaccompanied children, and pregnant women is recommended [42], however, this is not standardised for all facilities. This is partly caused by the fact that–

*"triage system is not standardised for all partners. Some use the Emergency Triage Assessment and Treatment (ETAT) protocol of WHO, some use the interagency integrated triage tool (IITT), and some do not even have any protocol. Standard protocols prioritise patients based on acuity or emergency; there is no mention of prioritisation on the grounds of older age, disability, or pregnancy. However, this is anyway happening in some facilities from the humanitarian ground…"* expressed by a triage/IPC expert.

One of the recommendations for protection mainstreaming in health response is infrastructural adaptations to the health facilities and restrooms, e.g., ramps and railings, handle grip, etc., for older persons and persons with disability [41,42]. According to the health sector quarterly monitoring report [47], less than 50% of the health facilities had a disability-friendly latrine. Only 39% of the health posts and 58% of the PHCs had the provision of ramps. Below half of the PHCs and only one-fourth of the health posts had the provision of side rails. Other measures placed by the health facilities for persons with disability included the provision of wheelchairs (74% health posts, 97% PHCs), stretchers (58% health posts and 86% PHCs), crutches (11% health posts, 23% PHCs) [47]. Therefore, a large number of health facilities are not using appropriate infrastructural and other measures to improve accessibility for persons with disabilities and older individuals. Some experts expressed that although some facilities placed some measures for the inclusion of persons with disability and older persons, in many cases, these are not functionalised. For instance, one of the experts shared that one health facility labelled a toilet as disability-friendly, but in practice, they were using it as a storeroom. Factors behind poor compliance of the health facilities for disability-friendly infrastructural adaptations include lack of sensitisation of the management and healthcare workers and lack of availability of infrastructural guidance.

While it is essential to ensure the labelling of the toilets is understandable to all literacy levels, universal icons/symbols are usually used as labels, which may not be comprehensible to the general community in the camps. Moreover, the designs of the toilets (e.g., high commode) and locking system may not be familiar to the beneficiaries. Therefore, it is recommended to consult with the community, including persons with disability and older people, on their preferences [42]. However, it is opined that such a consultation process merely happens in this context. One of the good practices mentioned is using Burmese or Rohingyalish (a Rohingya language written in the English alphabet) instead of English or Bangla for easy understanding by the refugee community.

One of the challenges mentioned was that toilets in many of the facilities are not friendly to children with or without disabilities. Although protection mainstreaming guidelines recommend that latrine design consider children, this aspect is often overlooked during installation. This could be caused by the program staff or engineers' lack of awareness of this issue. Similarly, most facilities do not provide separate space for children and lactating women, as is recommended; however, the majority of primary healthcare facilities maintain a breastfeeding corner.

Water, Sanitation and Hygiene Health Facility Improvement Tool (WASH FIT) outlined some indicators to monitor for gender and disability inclusion in WASH in health facilities, e.g., male-female segregated toilets for patients and staff, provision of menstrual hygiene management kits at least in one toilet and provision of disability-friendly toilet [49]. With WHO's initiative, a training program was cascaded among healthcare workers, facility managers, and engineers on this in 2019. Experts opined that this initiative significantly improved gender and disability inclusion in WASH at health facilities. The participants recommended continuing such training and placing appropriate follow-up and monitoring systems.

### Child protection

For the protection of children, the health facilities maintain close coordination with child protection actors in the camp. It is recommended that the waiting space should be child-friendly and there should be space for children to play [42]. Primary healthcare specialists who participated in this study expressed that this is often not possible to implement in a camp setting due to space limitations and overcrowding. However, many of the facilities have mental health and psychosocial units which organise entertainment and social events for children and adolescents. A few agencies provide space for children to play or engage in creative activities such as drawing. Participants expressed that there is a critical gap that training or learning opportunities for healthcare workers are not sufficient for child protection and communication with children. This results in a lack of initiative from the healthcare workers to create a child-friendly environment in the health facilities.

### Targeted interventions for vulnerable groups

There are only a few agencies in the camps that are implementing targeted interventions for persons with disability, older age and critically ill patients. Some agencies (e.g., YPSA, Helpage International) provide health screening, healthcare and referral support to geriatric patients through home-based care and age-friendly spaces. [50,51]. However, geriatric services are not mainstreamed with the provision of the primary healthcare provided by the PHCs and health posts. Few agencies, e.g., Humanity and Inclusion (HI), are providing specialised support to persons with disability with physiotherapy and assisted devices. Participants opined that such support to persons with disability is inadequate in comparison to the need. This finding is supported by an assessment conducted by Gender in Humanitarian Action Working Group, which reported that 56% of persons with disability reported not having received any assisted device in a year, as revealed by an assessment conducted by the age and disability working group [17]. The survey also reported that female older persons with disabilities were disproportionately affected; 67% of female older persons with disabilities had not received any assistive devices [17]. Further, participants of this study stated that there are no standardised guidelines or referral pathways available for the treatment, support and care of persons with disability and older people.

The referral system in Cox's Bazar, from the refugee camps to tertiary hospitals, mostly focuses on life-saving assistance to acutely ill patients. Patients with chronic conditions requiring long-term and cost-intensive treatment (e.g., chronic hepatitis, end-stage kidney diseases, advanced cancers) often do not get referral support for definitive treatment. Palliative care appears to be a new concept among health actors, and currently, only one organisation, namely IOM (International Organization for Migration), provides home- and facility-based palliative care in the camp.

*"There could be many patients in the camps who are at their terminal stage of disease. They may not need comprehensive treatment support. But at least they have the right to lead a peaceful life, physically and mentally, until the end of their life, meaning they will need palliation of their pain and suffering. However, palliative care is not mainstreamed with the primary healthcare service provision. As far as I know, IOM is the only agency that provides home-based and facility-based palliative care in a few camps. Such service needs further scale up".*

Although SRH and MHPSS are integrated into primary healthcare, very few interventions are taken by partners to improve the physical, mental, and psychosocial health of gender-diverse populations (GDP). It is believed that there remains a low level of awareness and a high degree of stigmatisation toward GDP.

## Prevention of and response to gender-based violence

Prevention of and response to Gender-based violence (GBV) is a multisectoral approach. Almost in all refugee camps, a comprehensive GBV referral pathway is established, mentioning the contact details of health (clinical management), mental health, case management (adult and children), legal aid and safe shelter service providers from the health and protection sectors [52]. Listed service providers are trained in providing relevant GBV services.

The Minimum Package of Essential Health Services for Primary Healthcare Facilities in FDMN (Forcibly Displaced Myanmar Nationals) Camps stipulates that primary healthcare facilities (24/7 clinics with inpatient) in refugee camps must provide first-line support, clinical care for sexual violence and intimate partner violence (i.e., Clinical Management of Rape – CMR) and facilitate safe referrals to other service providers [53]. While CMR is not a requirement for health posts (daytime outpatient posts), they should provide survivor-centred care, first-line support, and safe referral support. 93% of PHCs (n = 43) had the provision of GBV services, according to the quarterly monitoring report of the health sector [47]. More than 90% of these facilities offered emergency contraceptive pills and a CMR referral pathway, and approximately 70% provided post-exposure prophylaxis (PEP) and menstrual regulation kits [47]. Only 45% of health posts (n = 93) were found to have GBV health services, and only nearly half of these facilities had a CMR referral pathway [47]. The survey also revealed that three out of thirty-four refugee camps lacked a single facility that provides GBV post-exposure prophylaxis (PEP).

Research participants expressed one of the key challenges in CMR or GBV services at the facilities that there is a lack of awareness of the healthcare workers on the reporting arrangement; for example, many of the healthcare workers have the misperception that all cases of rape or sexual violence should be reported to the police. However, with repeated training and advocacy, this misperception is being significantly reduced among healthcare workers. Frequent staff turnover presents a challenge, requiring continuous training for the newly recruited staff. This also affects the availability of trainers in the pool for facilitating relevant training (e.g., CMR) since the lead agencies organise the training of trainers with the aim of creating a pool of trainers for cascading the training among healthcare workers at health facilities.

Some agencies are finding better ways to integrate health and protection support for GBV survivors. For example, some agencies have GBV case workers within the health facility, which can provide integrated health and protection services. Another best practice noted by an agency is having healthcare workers (e.g., midwives) in the women and girl safe spaces (WGSS) and secure and health learning spaces for providing health care and education services in integration with GBV prevention and case management services of the protection sector. Several agencies engage their CHWs to raise awareness of the community on GBV and its referral pathways, enhancing access of GBV survivors to care and support.

## Communication with community and accountability to affected population

Engagement of community including all layers of the community, including women, men, girls, boys, children, elderly, GDP and persons with disability throughout the project life cycle is an essential requirement for accountability to the affected population as well as to appeal to Joint Response Plan [42,43]. Participants expressed that community engagement in health program design is often neglected. There is no mechanism in place to track whether actual consultation with the community was made during the project design. Some participants also emphasised that there is limited effort to engage with women's groups, and most community groups are dominated by male community leaders (e.g., Majhi/Imams).

The establishment of a community feedback mechanism is another area of focus for AAP [43]. Participants expressed that most facilities have some form of community feedback mechanism in place, e.g., feedback box, hotline number, box and coin method. Participants expressed that these mechanisms are not often utilised because either participants may be unaware of the system or the system may not be flexible to all, e.g., illiterate, children, persons with disability. In many cases, it is also unclear how or whether these systems can influence the improvement or adjustment of the quality of service. Participants mentioned a good practice where some health facilities are facilitating community health facility support

groups. These groups, which include representatives from different gender and ages, persons with disability and community leaders, serve as advisory bodies and provide feedback to facility managers through periodic meetings. The input allows the management to take necessary measures to improve its service.

There is a network of 1300 community health workers (CHW) across the refugee camps who perform risk communication and community engagement activities. Most of them are recruited from the local community (i.e., Rohingya), and the majority of them are women. Being women, they have more accessibility to the households and can discuss things with female family members. CHWs facilitate health promotion sessions, perform active case searches and referrals, follow-up immunisation, and raise awareness of gender and various health issues in the community through regular household visits and community group meetings. People from different gender, age groups and background participate in these sessions.

Several working groups and agencies are active in this setting for the development and dissemination of information, education and communication (IEC) materials, including flip charts, posters, audio clips and videos in local languages. Some participants suggested that the community should be more engaged in developing risk communication strategies and materials. Concerns of different gender, age groups and persons with disability are often not considered in many communication products and strategies. As one participant questioned -

*"We have developed frequently asked question aids for dengue risk communication. CHWs use this job aid to share the information during their household visits or group sessions. But have we considered any culturally familiar methods, e.g., street drama or folk songs? Have we considered how we will reach out to those messengers who speech and hearing impairments? How about older people who often cannot join during the sessions?"*

## Information management

One of the core requirements for gender analysis of health is generating and reporting cases of COVID-19 and OPD (Outpatient Department) services with sex, age and disability disaggregation [41]. Multiple reporting systems are in place in Cox's Bazar for disease surveillance and health information management. While disaggregation of male-female data is happening in all reporting systems, some key reporting systems have limitations for age and disability disaggregation. Most of the systems do not have segregation options for gender-diverse populations. WHO Early Warning, Alert and Response System (EWARS) is a surveillance tool designed for early detection and warning of infectious disease outbreaks. The weekly indicator-based surveillance platform of the system collects and reports data disaggregated by sex (male and female) and age group over-5 and under-5 years. District Health Information System −2 (DHIS-2) also have a similar arrangement for daily morbidity reporting. Therefore, both systems fail to give analysis regarding the accessibility, morbidity and concerns of different age groups, e.g., older persons, adolescents, and persons with disability. However, EWARS and GoData (another surveillance tool of WHO) have case report forms for some ongoing/threatening outbreak conditions, e.g., COVID-19, Diphtheria, Dengue, acute watery diarrhoea, Acute Flaccid Paralysis, which collect detailed case-by-case information, including exact age of the individuals, allowing proper segregation of data by age group. Yet, disability disaggregation is not included in these systems. Only a few agencies use the Washington Group Short Set on Functioning (WG-SS) tool as part of their internal health information system to document the proportion of individuals treated at their health facilities who have a disability. However, experts have noted that this tool has limitations in detecting certain conditions in children below two years of age. This aligns with broader pediatric diagnostic challenges, where some disabilities may not be evident until later developmental stages.

## Emergency preparedness and response

Cox's Bazar is a disaster-prone district, and the Rohingya refugees and adjacent host communities are highly susceptible to the direct and indirect health impacts of cyclone and monsoon-related heavy rains and landslides, such as drowning,

mass casualty, and outbreaks of disease [54]. Mobile medical teams play a crucial role in such preparedness and response. The Mobile Medical Team's (MMTs) are required to have a midwife to ensure the integration of SRH services and the representation of female personnel. The MMT team composition includes a protection officer who leads the identification of people requiring protection assistance, such as separated families, unaccompanied children, survivors of GBV, persons with disability, and older people, and facilitates the referral of the identified individuals to protection service providers (such as protection focal persons, Protection Emergency Response Unit – PERU) [55]. Partners have various options for engaging a protection officer in the team, including a) having a dedicated protection staff from the agency, b) having a seconded protection sector staff within the MMT, and c) training MHPSS staff or other team members to provide protection services. Protection officers are trained to provide survivor-centre support to GBV survivors, including psychological first aid (PFA) and safe referrals; to provide child-friendly support to unaccompanied and separated children (UASC) or children who have experienced any form of violence, and to facilitate referrals of vulnerable individuals in need of specialised protection assistance.

The Mobile Medical Team Working Group of the health sector conducts regular training on protection mainstreaming for MMT health staff and protection officers. This is another example of a best practice for protection mainstreaming in emergency health response. A functional link is also established between Mobile Medical Teams and Protection Emergency Response Units for referring survivors of gender-based violence to each other for clinical management and protection assistance, respectively. However, experts in emergency health response believe that the central coordination between the health and protection sectors should be strengthened to standardise the mainstreaming of protection focal persons across all MMTs.

### COVID-19 outbreak response

The COVID-19 outbreak within the camps and the pandemic as a whole had a significant impact on gender and vulnerabilities. Humanitarian partners have taken substantial steps to mainstream gender and protection into COVID-19 preparedness and response. One of the indicators set in the health sector GAP is "Percentage of reports of FDMN/Rohingya refugee and host community samples tested with results disaggregated by age and sex" [41]. In the case management system, partners had the provision of separate male, female and child wards with proper privacy.

One challenge the healthcare workers and the general community encountered was that many personal protective equipment (PPE) were not gender and child-inclusive. For instance, the masks provided to the community or health facility were not designed for children. Outbreak response experts informed that the gowns and scrubs were not designed according to the body structure of women. However, there were also some best practices mentioned for gender mainstreaming; for instance, one of the partners produced and distributed "medical hijab' and culturally friendly scrubs to women.

One of the misconceptions persists within the community that if the women wear *nikab* (a thin piece of garment that cover face of Muslim women), they do not need to wear a mask assuming *nikab* and mask give same level of protection. As one of the experts expressed, *"…the perception that nikab can protect women from COVID-19 increased their exposure to the virus and as a result proportion of COVID-19 among women gradually increased".*

The health sector and child protection subsector developed comprehensive guidance for the care of children in health facilities during COVID-19 [44]. From each Severe Acute Respiratory Illness Isolation and Treatment Center (SARI ITC), the child protection actors trained some healthcare workers on psychological first aid (PFA), psychosocial support, and communication with children. These trained healthcare workers acted as "child carers" to provide care to unaccompanied children while in the centre. The SARI ITCs were linked with child protection actors to address a variety of child protection concerns, including a) how to mitigate if a child leaves the centre with a non-caregiver if the child is distressed seeing critically ill patients, b) who will accompany a child with COVID-19 at a SARI ITC if the formerly healthy caregiver contract COVID-19, c) what alternative options are available for healthy children of a COVID-19 infected caregiver, d) how the child

will be supported if their caregiver dies at the ITC, e) what support can be provided to the children abandoned at the ITC or f) how to mitigate the risk of children/adolescents running away from the health facility. However, in practice, many partners faced challenges in admitting a COVID-19-infected caregiver who might have multiple dependent young children.

Participants in the study view the COVID-19 vaccination campaign for Rohingya refugees in Cox's Bazar as one of the best examples of gender and protection integration in a health response. The vaccination campaign began with individuals over 55 years old and expanded to include all adults (≥ 18 years) and adolescents over time (12–17 years). During this study, preparations were made to vaccinate children aged 5–11. Through the network of CHWs and volunteers in communication for development, extensive risk communication efforts were made to mobilise all target populations, including men, women, persons with disability, and the elderly. Some health agencies collaborated with site management and protection actors to identify individuals with mobility issues and provide porter assistance (e.g., persons with disability, extreme age, critically ill patients).

## Discussion

This study provides critical insights into gender and protection mainstreaming in humanitarian health response, highlighting how self-efficacy among professionals influences the success of these efforts. Strengthening leadership capacity, particularly for women, is essential for sustainable progress. To the author's knowledge, this is the first study ever conducted in a humanitarian setting on the practice of gender and protection mainstreaming in health response. The discussion is organised into eleven broad themes, the same as the result section.

### Strategy, partnership and collaboration

Health and protection sectors in Cox's Bazar developed a significant number of gender and protection mainstreaming plans and guidelines. The requirement for the application of Gender with Age Marker (GAM) for gender and protection mainstreaming in making JRP appeal is one of the best practices in this setting. Another key strength identified in this setting is the extensive collaboration of the health sector with the protection sector, child and GBV subsectors and GiHA. While progress has been made in gender inclusion, the absence of leadership pathways affects women's self-efficacy in assuming managerial roles. Leadership training and mentorship could enhance their confidence and participation in decision-making [56]. This gap is noted not only for this particular context in Cox's Bazar but also found in other gender and protection mainstreaming plans/strategies [19,20]. Further, although the health sector strategic plan recognises the need for a minimum standard of disability inclusion for health response, unlike gender mainstreaming, the coordination mechanism is less focused on addressing other vulnerabilities, especially persons with disability and older age individuals. The sector has a system in place to monitor the adherence of the partners to different recommended measures, however, the system focuses mainly on quantitative achievement rather than functionality and quality. The system can be further strengthened to address this concern by adding qualitative indicators and methods into the M&E system [56].

### Staffing and capacity building

One of the key strengths of this context is the significant representation of women in the health workforce, as evidenced by the fact that all health facilities have at least one female medical professional staff member. The study revealed, however, that the distribution of female employees is not proportional across all positions and that women are underrepresented in leadership and coordination roles, particularly in program management, camp coordination, and leading the working groups. The findings of this study are similar to studies in some other contexts. For instance, Patel et al. identified that women are underrepresented in humanitarian leadership [23], and Witter et al. found that women were largely employed in nursing and midwifery positions and under-represented in management [25]. Cultural and social norms impact women's self-efficacy in humanitarian roles. Targeted mentorship and leadership initiatives can enhance their

confidence and career persistence [57]. The fact that there is no special prioritisation system in place for women to be in leadership roles. This is similar to the finding of RedR in Jordan that female staff have to balance their career and cultural expectations of household activities [26].

The study also found that although training is organised on different aspects of gender and protection mainstreaming, there is a lack of focus on training on child protection, communication with children and inclusion of various vulnerable groups, e.g., older persons, persons with disability and people, and people living with HIV. Also, the training opportunities are not equal for all agencies, and sometimes training methods are not adapted according to the capacity level of the staff.

## Safety and security

A strength of this response is that the health sector and partners conduct a rationalisation exercise that can recommend partners to establish or relocate their facilities to a safer and more accessible location. This appears to be an effective strategy given that some partners have relocated their facilities to safer places following the rationalisation's recommendation. Even though natural disasters such as monsoons and cyclones pose critical threats to health operations, the health partners implement mitigating measures, such as the operation of mobile medical teams, to provide lifesaving health services. The installation of general security measures, such as lights, security guards, etc., by most facilities is also recognised as an effective safety measure. Unstable power supply increases GBV risks and limits staff confidence in emergency response. Strengthening infrastructure could enhance both service efficiency and staff self-efficacy [58]. A sustainable emergency solution should be planned at the sector and policy level, taking into account different best stories in a similar context [57,58].

The placement of infection and control measures in most health facilities by the partners is a strength of this system to prevent hospital-acquired infection among staff, patients and the community. However, further science-based research should be undertaken to better understand the effectiveness of the protocols in the facilities [59]. The study found that Cox's Bazar is yet to have a comprehensive biomedical waste management system which is linked to the protection risk of the community and healthcare workers due to the infectious and hazardous nature of biomedical waste. This concern is not specific to Cox's Bazar but generalised for other emergency and low-resource settings, exacerbated during the COVID-19 pandemic [60,61]. The health partners should coordinate and share responsibilities for a harmonised waste management system exploring innovative solutions [60].

## Infrastructure, WASH and patient flow

This study found a mixed type of adherence of partners to the recommendations on infrastructure, WASH and patient flow in relation to gender and protection mainstreaming. While most of the facilities have sex-segregated toilets, only half of the facilities have a disability-friendly toilet, and consideration of access to children is merely considered in the toilet facilities. Many health facilities lack proper infrastructure for older persons and persons with disabilities, reducing their accessibility and protection. Context-specific WASH guidelines are needed to standardize gender and disability inclusion. Also, concern was raised that having the physical presence of these segregated toilets or adapted structures does not guarantee their appropriate utilisation by the target groups. These concerns could be linked to a lack of guidance on infrastructure design, a lack of sensitisation of facility designers on gender and protection mainstreaming, as well as a lack of orientation of the healthcare workers on the need for such segregation and arrangement. Having separate consultation rooms for males and females is often not practical in many facilities, especially in small facilities with limited human resources.

Although the Water, Sanitation, and Hygiene Health Facility Improvement Tool (WASH FIT) outlines some indicators to monitor gender and disability inclusion in WASH in health facilities, comprehensive guidance should be developed for the partners to address the above concerns of gender and protection mainstreaming in health facility infrastructure, WASH, and patient flow systems.

Prioritisation of older people, persons with disability, unaccompanied children and pregnant women in the patient queue is not standardised and not practised universally in all facilities. This is linked to a lack of standardised triage tools for all health facilities and the fact that standard triage tools used globally (e.g., ETAT, IITT) prioritise patients based on acuity, not based on social vulnerability. The study also found that vulnerable groups with protection concerns (e.g., women, pregnant and lactating mothers, older persons, persons with disability, and people living with HIV) are often not consulted in practice to learn their concerns, needs and cultures, which could be reflected in the health facility design and infrastructural adaptation.

### Child protection

The research identified a critical gap in the capacity building of healthcare workers on communication with children and child protection. Strengthening the child protection system is crucial to preventing and responding to abuse and exploitation [62] and strengthening the health workforce can contribute to strengthening the child protection system. The research also identified some best practices, e.g., organising social/entertainment events for children and having space for children to play/art by a few agencies, which can be scaled up for better psychosocial well-being of the children.

### Targeted interventions toward vulnerable groups

The research found that only a few agencies in the camps are engaged in rehabilitative, palliative and home-based care for persons with disability and older persons. Also, there are no standardised guidelines or referral pathways made available for the treatment, support and care of persons with disability and geriatric illness. This can be related to the fact that rehabilitative and palliative care are not integrated with the primary healthcare provision in Cox's Bazar [53] and there is limited data available on the barriers and concerns of persons with disability and older people not only in Cox's Bazar but globally in other contexts [10,33]. A lack of targeted initiatives for the gender-diverse population (GDP) limits inclusivity efforts. Developing specialized interventions can reduce systemic inequalities and improve service accessibility. Therefore, evidence-based targeted initiatives are warranted to address the specific barriers and opportunities of people with different vulnerabilities.

### Prevention and response to gender based violence (GBV)

The study found that a significant effort has been made to prevent and respond to GBV in the Rohingya refugee camps, as evidenced by a comprehensive GBV referral pathway in place for all camps, and over 90% of the PHCs have the provision of GBV services. Despite strong GBV response mechanisms, gaps remain in CMR referral pathways and GBV PEP service availability. Strengthening coordination can enhance service access and timeliness. The study also found some key challenges, like misconceptions among staff regarding reporting GBV cases and frequent staff turnover affecting the availability of trainers for facilitating relevant training. One of the best practices identified by the research is that some agencies provide GBV medical (CMR) and protection (case management) services in the same facility, enhancing the timeliness, promptness, effectiveness, and confidentiality of the response. Another innovation is that an agency has healthcare workers (e.g., midwives) in the women and girls safe spaces (WGSS) and safe and health learning spaces to provide healthcare and education services integrated with GBV prevention and case management services. With the partners' efforts and a strong monitoring and coordination system, GBV service is well-mainstreamed in health response. This is significant progress in comparison to the preliminary situation in refugee influx, when a study found that the public health and aid agencies gave little attention to the survivors of GBV in Rohingya refugee camps [63]. The progress can be considered as a strength in this response, comparing the findings with other similar settings like the Uganda crisis, where the GBV response faced several challenges, including the inadequate provision of screening, health staff, drugs and treatment psychosocial counselling [64,65].

## Communication with the community and accountability to the affected population

Our study found that although engaging different layers of the community – including women, men, girls, boys, children, elderly, gender-diverse persons (GDP) and persons with disability throughout the project life cycle is a principle recommended by the humanitarian sector, its practical implementation often remains inconsistent and is frequently neglected in practice.

Partners have various methods of collecting feedback from the community, but concern was raised on whether such feedback is taken into consideration to influence the facility's quality improvement. However, one of the best practices mentioned is having a community health facility support group with representation from different layers of society, which can have direct dialogue with the facility management and influence system change and improvement. The research also found that a network of 1300 community health workers, mostly female, play a critical role in community engagement through household visits and community sessions. These findings are similar to another research study by the author on COVID-19 risk communication and community engagement, which concluded that the selection of CHWs from the local community, deployment of female CHWs, and speaking in the local dialect can significantly contribute to community engagement [66]. Engagement of female CHWs was also considered best practice in other contexts [30,67]. However, the risk communication and community engagement strategy should be more inclusive of gender, age, and disability, and specific strategies and materials should be designed to reach out to different groups. WHO community engagement guidelines also emphasise local understanding and engagement consistent with the language, culture, and context [68].

## Information management

Although one of the core requirements for gender and protection mainstreaming is to have sex, age and disability disaggregated data, some key reporting systems in Cox's Bazar, e.g., EWARS, DHIS-2 only desegregate data based on sex (male-female) and below five years and over five years with the exception for case report forms for ongoing outbreaks having details of gender and age. The systems hardly have options for disaggregation of data for persons with disability and GDP. The finding of this study echoes the result of a study by Robinson el (2020) [10], which found limited evidence on data collection and identification of persons with disability and older persons in humanitarian response. Disaggregation of data to the extent possible enables to identify whether assistance is being distributed or accessed impartially or disproportionately [2]. Therefore, the reporting systems should be upgraded to have proper disaggregation based on age (inclusive of adolescent and older age), gender (not just sex) and disability. The Washington Group Short Set (WG-SS) is a useful tool for disability data collection, but its lack of child inclusivity requires complementary tools like DTM-MSLA for better representation. Therefore, WG-SS should be further scaled up at all facilities and other relevant tools, e.g., DTM MSLA for Disability Inclusion, can be used to identify barriers faced by persons with disability [69,70]. Washington Group/ UNICEF child functioning module can be used for surveying children with disability [71].

## Emergency preparedness and response

Health and protection partners demonstrate strong collaboration in Cox's Bazar's emergency response. Integrating protection focal persons in emergency teams enhances service efficiency. During emergencies, a protection officer remains as a part of the emergency mobile medical teams to deal with the protection concern. Mobile medical teams also coordinate with Protection Emergency Response Units (PERU) to integrate the management of protection concerns (e.g., GBV, UASC). Additionally, the mobile medical teams ensure that midwives and MHPSS staff in the teams provide SRH, mental health, and psychosocial support. Our study echoes the finding of an After Action Review following a massive fire incident in 2021, which highlighted the critical role of a dedicated protection focal person within the Mobile Medical Team (MMT) in mainstreaming protection during emergency health responses. The focal person was responsible for identifying individuals with specific protection needs, facilitating immediate referrals for gender-based violence (GBV) survivors, ensuring

child protection interventions, and coordinating with humanitarian actors to provide psychosocial and medical support. This case underscores the importance of embedding protection personnel within emergency health teams to enhance response effectiveness in similar crisis settings [72].

**COVID-19 outbreak response**

The COVID-19 response in Cox's Bazar made some remarkable arrangements for gender and protection mainstreaming, which includes age and sex-disaggregated data systems and separate male, female and child wards. There was strong collaboration between health and child protection partners for admission of children with COVID-19 or caring for children with a COVID-19-positive caregiver. The Child Protection subsector also trained healthcare workers as "child carers" to provide for unaccompanied children while in the centre. A key challenge was the lack of gender-inclusive PPE. Culturally appropriate adaptations, such as medical hijabs, improved usability and acceptance. This is not just a local concern in Cox's Bazar but a general concern that happened globally [72]. However, partners demonstrated some innovative solutions of having "medical hijab' and culturally friendly scrubs for women, which can be marked as a best practice and replicated in other outbreaks. The study also found that there was confusion or misconception among women regarding the *nikab*, whether it is equivalent to a mask. Such confusion was also found in other settings in the world [73]. However, this confusion/misconception was not addressed in the risk communication and community engagement strategy, which increases the exposure risk of women to COVID-19. The COVID-19 vaccination campaign for the Rohingya refugees is considered another best example of gender and protection mainstreaming where all gender and age groups are gradually enrolled. A best practice of collaboration among health, site management and protection actors was found to identify and provide porter support to individuals having mobility issues (e.g., persons with disability, extreme age, and critically ill patients).

**Limitation of the study**

This research could be more informative if more data collection methods were employed, e.g., actual observation of gender and mainstreaming processes in the refugee camps, focus group discussion, or interviews with vulnerable groups. However, due to the limitations of time and resources and consideration of protection concerns, the research methods were limited to the literature review and in-depth interviews of professionals. Since data analysis was conducted by a single researcher, thematic coding bias is a potential limitation. To mitigate this, the study cross-referenced findings with established frameworks and existing policies. Additionally, a secondary researcher with qualitative expertise reviewed key themes to enhance analytical rigor. To minimize bias, exclusion criteria were applied, ensuring that only responses directly relevant to the study's scope were included. Responses that lacked depth or significantly diverged from the study focus were excluded. While this approach ensured thematic consistency, it may have limited the diversity of perspectives. While qualitative methodologies allow for in-depth understanding, the study lacks statistical generalizability. Future research should incorporate participatory methods, including direct engagement with refugee populations, to provide a more comprehensive understanding of the issue. Moreover, employing investigator triangulation or peer validation could enhance the reliability of qualitative findings. Sexual and reproductive health concerns were intentionally excluded from the study, considering the availability of extensive literature in this field.

**Conclusion and recommendations**

Gender and protection mainstreaming in humanitarian health response is a complex phenomenon that involves multiple sectors and actors from all layers. While this study highlights key interventions, it does not claim to provide an exhaustive framework. Instead, it offers a structured analysis of current practices and gaps, serving as a foundation for further targeted research and policy refinement. We acknowledge that these findings are context-specific to the Rohingya refugee crisis in Cox's Bazar and may not be generalizable to other humanitarian settings without further research.

 

Analysing existing strategies, tools and guidelines, our study identified 68 recommended interventions under nine general themes, i.e., partnership and coordination, safety and security, infrastructure, WASH and patient flow, child protection measures, targeted interventions toward vulnerable groups, GBV prevention and response and information management. Exploring practices on gender and protection mainstreaming, we added two more themes: emergency preparedness and response and COVID-19 response. The study revealed a range of good practices on gender and protection mainstreaming in health response, e.g., placement of a gender action plan, application of Gender with Age Marker (GAM), monitoring system for gender and disability inclusion, rationalisation exercise on location and distribution of health facilities, emergency preparedness and response system, infection prevention and control system, availability of sex-segregated toilets and waiting spaces, established GBV referral pathway and availability of GBV service at most PHCs and engagement of female CHWs from local community. The study also revealed some best practices that are happening on a small scale but could be scaled up, such as arranging psychosocial events/spaces at health facilities for children, palliative care for terminally ill patients, integrated medical and protection services at health facilities, and facilitating community health facility support groups. We identified some interventions that need further improvement or scale-up, e.g., the provision of disability-segregated toilets, setting up a comprehensive waste management system, provision of sex, age and disability-disaggregated data and a responsive community feedback mechanism. We also found critical gaps in some areas, e.g., lack of women's leadership in health response, gaps in terms of coordination, capacity building, strategy planning and targeted interventions to address needs and concerns of older age, disability, GDP and other vulnerable groups, limited scope of monitoring the quality/functionality of the interventions, lack of adherence to infrastructural adaptation measures, inconsistent power supply to the health facility, limited effort in consultation with community, especially women and vulnerable groups on their concerns and absence of any triage protocols that prioritise socially vulnerable individuals.

Based on the research findings, we recommend the following actions to improve gender and protection mainstreaming in health response.

(1) There should be a comprehensive gender and protection mainstreaming practical guide for the partners harmonising all existing resources with details on practical solutions. The focus should be given to the inclusion of all vulnerabilities, including gender, age, disability and any form of discrimination and stigmatisation. In line with the guideline, the Gender Action Plan can be further expanded, covering all vulnerabilities with clear actions.

(2) Women's leadership in health response should be promoted and strategised, ensuring their representation in all technical, management, and coordination roles at all levels.

(3) The Monitoring and evaluation system for gender and protection mainstreaming should be further strengthened, incorporating qualitative measures to measure the quality and functionality of the interventions. A mixed-method monitoring approach, integrating qualitative insights with quantitative tracking, should be considered to ensure a more comprehensive evaluation of gender and protection interventions (Reference: Section 4.3 on Monitoring Gaps).

(4) A comprehensive capacity-building plan on gender and protection issues should be developed, ensuring the benefit of staff from all levels of agencies and staff.

(5) All facilities in inaccessible areas or with unsafe routes should be relocated to safer and more accessible locations. Decisions on facility relocation should be guided by accessibility mapping and community consultation to ensure that new locations effectively serve vulnerable populations (Reference: Section 4.2 on Community Engagement Barriers).

(6) Advocacy initiatives should be undertaken to ensure consistent power supply to health facilities. Low-cost innovative solutions can be further researched.

(7) The effectiveness of infection prevention and control (IPC) protocols should be researched, and IPC measures (including PPE) should be gender-inclusive.

(8)   Infrastructural and WASH guidance should be provided to the health partners and relevant engineering teams, including gender, age, and disability, with a proper monitoring system in place. Incorporating participatory design principles, particularly through consultations with women, persons with disabilities, and older persons, can enhance the inclusivity of WASH and infrastructure interventions (Reference: Section 4.1 on Inclusivity Challenges).

(9)   Triage protocols should be upgraded to prioritise socially vulnerable individuals.

(10)   Child protection in health response, including the provision of a child-friendly environment and communication, should be further strengthened through better collaboration between health and child protection partners. The existing best practices of psychosocial support for children should be promoted and scaled up.

(11)   Targeted interventions towards neglected groups, e.g., persons with disability, older persons and GDP, should be undertaken, scaled up and included in the Joint Response Plan. There should be established referral pathways for such services.

(12)   Existing innovations and best practices for integrated GBV prevention and response interventions should be promoted, included in the strategy and further scaled up. Health posts should be further capacitated on the GBV referral pathways.

(13)   Engagement with the community and AAP, including all vulnerable groups, should be intensified through the implementation and monitoring of the existing AAP plan.

(14)   Risk communication and community engagement strategies should be responsive to the needs of all genders, ages, disabilities, and vulnerable groups, considering local culture, context, and language.

(15)   All surveillance and reporting tools should have measures for collecting gender (not only sex), age (all groups), and disability disaggregated data. A system should be in place to identify barriers to vulnerable groups.

(16)   The best practices of COVID-19 response (e.g., an age- and disability-inclusive vaccination campaign and child protection in SARI ITCs) can be replicated in other outbreak settings, such as dengue and diphtheria.

We urge policymakers, sector coordinators, health program managers, and healthcare professionals to address the aforementioned gaps and challenges, learn and scale up the best practices, and apply the recommendations mentioned above. While this study focuses on gender and protection mainstreaming in a specific humanitarian context, its findings may offer insights for similar crisis settings. However, further localized research is required to assess contextual variations and applicability in different humanitarian responses. Future research should build upon these findings by employing a broader mixed-method approach, integrating perspectives from directly affected populations. Expanding the scope to include long-term impact assessments and policy implementation effectiveness will further contribute to strengthening gender and protection mainstreaming in humanitarian health responses. Although this research focused on gender and protection mainstreaming at the local level, many of its findings are applicable to other humanitarian contexts. Consequently, global health and protection clusters should consider integrating these insights to strengthen gender-responsive and inclusive health interventions in crisis settings. Moreover, this study paved the way for future quantitative and qualitative research on the themes and actions found in this study pertaining to gender and protection mainstreaming in humanitarian health responses.

## Supporting information

**S1 File.   Guiding questions for key informant interviews.**
(DOCX)

**S2 File.   Thematic analysis (Qualitative dataset).**
(XLSX)

## Acknowledgments

This article is based on a dissertation by the author that was submitted to the University of Manchester for the MSc in Global Health at the Humanitarian and Conflict Response Institute in the Faculty School of Humanities. The author thanks everyone who provided their generous support for the successful completion of this study and acknowledges the contribution of the 12 health and protection experts in Cox's Bazar who took part in the in-depth interviews and gave their valuable time, information and opinions for this research.

## Author contributions

**Conceptualization:** Charls Erik Halder.

**Data curation:** Charls Erik Halder, Md Abeed Hasan.

**Formal analysis:** Charls Erik Halder.

**Investigation:** Charls Erik Halder.

**Methodology:** Charls Erik Halder.

**Project administration:** Charls Erik Halder.

**Resources:** Charls Erik Halder, Md Abeed Hasan.

**Supervision:** Charls Erik Halder.

**Validation:** Charls Erik Halder, Md Abeed Hasan.

**Visualization:** Charls Erik Halder.

**Writing – original draft:** Charls Erik Halder.

**Writing – review & editing:** Charls Erik Halder, Md Abeed Hasan.

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
