## [Decision Letter · Decision Letter 0]

Dear Dr. Halder,

Thank you for submitting your manuscript to PLOS ONE. After careful consideration, we feel that it has merit but does not fully meet PLOS ONE’s publication criteria as it currently stands. Therefore, we invite you to submit a revised version of the manuscript that addresses the points raised during the review process.

We look forward to receiving your revised manuscript.

Kind regards,

Md. Feroz Kabir, BPT, MPT, MPH, BPED, MPED

Academic Editor

PLOS ONE

Journal Requirements:

1. When submitting your revision, we need you to address these additional requirements. Please ensure that your manuscript meets PLOS ONE's style requirements, including those for file naming. The PLOS ONE style templates can be found at https://journals.plos.org/plosone/s/file?id=wjVg/PLOSOne_formatting_sample_main_body.pdf and https://journals.plos.org/plosone/s/file?id=ba62/PLOSOne_formatting_sample_title_authors_affiliations.pdf 2. We note that your Data Availability Statement is currently as follows: [All relevant data are within the manuscript and its Supporting Information files.] Please confirm at this time whether or not your submission contains all raw data required to replicate the results of your study. Authors must share the “minimal data set” for their submission. PLOS defines the minimal data set to consist of the data required to replicate all study findings reported in the article, as well as related metadata and methods (https://journals.plos.org/plosone/s/data-availability#loc-minimal-data-set-definition). For example, authors should submit the following data: - The values behind the means, standard deviations and other measures reported;- The values used to build graphs;- The points extracted from images for analysis. Authors do not need to submit their entire data set if only a portion of the data was used in the reported study. If your submission does not contain these data, please either upload them as Supporting Information files or deposit them to a stable, public repository and provide us with the relevant URLs, DOIs, or accession numbers. For a list of recommended repositories, please see https://journals.plos.org/plosone/s/recommended-repositories. If there are ethical or legal restrictions on sharing a de-identified data set, please explain them in detail (e.g., data contain potentially sensitive information, data are owned by a third-party organization, etc.) and who has imposed them (e.g., an ethics committee). Please also provide contact information for a data access committee, ethics committee, or other institutional body to which data requests may be sent. If data are owned by a third party, please indicate how others may request data access. 3. Your ethics statement should only appear in the Methods section of your manuscript. If your ethics statement is written in any section besides the Methods, please move it to the Methods section and delete it from any other section. Please ensure that your ethics statement is included in your manuscript, as the ethics statement entered into the online submission form will not be published alongside your manuscript.

Additional Editor Comments:

Submit your revised manuscript according to the reviewers comments within the next 14 days.

Reviewers' comments:

Reviewer's Responses to Questions

**Comments to the Author**

1. Is the manuscript technically sound, and do the data support the conclusions?

Reviewer #1: Partly

Reviewer #2: No

Reviewer #3: Yes

2. Has the statistical analysis been performed appropriately and rigorously?

Reviewer #1: No

Reviewer #2: N/A

Reviewer #3: N/A

3. Have the authors made all data underlying the findings in their manuscript fully available?

Reviewer #1: Yes

Reviewer #2: Yes

Reviewer #3: No

4. Is the manuscript presented in an intelligible fashion and written in standard English?

Reviewer #1: Yes

Reviewer #2: No

Reviewer #3: No

Reviewer #1: Thank you for this opportunity to revise the manuscript titled “The Practice of Gender and Protection Mainstreaming in Health Response in Humanitarian Crisis - A Case Study from the Refugee Camps in Cox’s Bazar, Bangladesh” that was submitted to PLOS ONE.

I would like to extend my thanks to the authors for their work on the manuscript and for undertaking such an important topic. The results are definitely worth publishing and I am also looking forward to the future plans mentioned in the text coming to fruition. However, the article itself needs a significant amount of work to be ready for publication. I outline the issues that caught my attention.

I hope those are helpful for the authors in improving this important and timely work.

1. Provide more information on the process of obtaining consent from the participants and how confidentiality and anonymity were ensured. This needs more elaboration.

2. If a student needed emotional support after completing the questionnaire was this available and indicated on the information letter?

3. While, judging by sample size there should be no concerns regarding statistical power, additional comments about power considerations would be valuable.

4. Given the properties of the analytic procedure, please consider including the additional information on potential deviations from univariate and multivariate normal distribution.

5. Your discussion is too descriptive in my view. The resulting findings look very general. You mainly summarize the results, but you don't discuss the broader implications of these findings for self-efficacy.

6. After revising the manuscript, please make sure the abstract accurately summarizes it.

I will be glad to review the revised manuscript.

Reviewer #2: I would like to provide my feedback on the manuscript, which presents significant concerns regarding its structure and clarity.

1. Title and Study Design: The title and study design do not align appropriately. It is unclear why this study qualifies as a case study rather than a qualitative study, particularly one based on in-depth interviews. This discrepancy needs to be addressed to provide clarity to the readers.

2. Terminology: The term “Gender and Protection Mainstreaming” may not be commonly recognized. As a reader unfamiliar with this concept, I found it difficult to understand its implications after reading both the abstract and the background section. A more appropriate term, such as “gender and vulnerable population policy adherence” or “gender protection policy adherence,” could enhance clarity.

3. Methods Section: The methods section in the abstract is insufficient. It omits critical information about who the “professionals” are and how the data was analyzed. Providing this context is essential for understanding the methodology's validity. Further in the main manuscript, it is unclear whether the data was analyzed solely by the author or someone else. If solely by the author, significant bias can be introduced that needs to be mentioned in the limitation of the study.

4. Results Section: The results section lacks detail regarding the participants. It would be beneficial to explain the rationale behind selecting this sample. A basic demographic statistics explaining their background should also be included. Additionally, the results presented are overly broad and do not align with the focused nature typically expected from a case study. If a qualitative study, I can understand the themes better.

5. Conclusion: The conclusion is overly ambitious. It should be reframed to more accurately reflect the study's findings and their implications, avoiding grand statements that may not be justified by the data.

6. Overall Clarity: The manuscript, while containing relevant content, is disorienting due to its lack of clear definitions and context. The background section should explicitly define “gender and protection mainstreaming” and clarify why a case study is being conducted. Without proper orientation in the abstract and background sections, the remainder of the manuscript is challenging to follow and comprehend. I recommend addressing these issues to enhance the manuscript's clarity and coherence.

Reviewer #3: I want to thank the author for this interesting manuscript. It is certainly insightful and supportive to the gender mainstreaming and humanitarian response in camp/emergency settings and beyond. I particularly appreciated the concise discussion, conclusions and recommendations provided – which will certainly be valuable to actors in Cox’s Bazar and similar contexts.

I am less familiar with the method chosen, yet did my best to contribute to improvement of the work.

Re question 3, I am uncertain if PLOS requires qualitative data to be shared, which I would doubt. I clicked NO, as no data set is made available, yet I assume this is not needed in this case.

Re question 4, I would advise for a thorough review and editing process, reducing repetition and increasing concise and clear language. Particularly the result section could be improved. The discussion and conclusion sections were really good.

I am providing very detailed comments with the aim to improve the manuscript and its contribution to the field.

Further proof reading and support with condensing could be useful. At times sentences are very long, and could be broken up into 2 or more to aid understanding.

Introduction

Persons with disability and older persons often face discrimination based on their age and disability. -> switch age and disability at the end to match the order of the start of the sentence.

To meet the demand currently, 80 health partners are providing primary and secondary healthcare services in the camps through 93 health posts, 43 Primary Health Care centres (PHCs), Field Hospitals and a network of 1300 community healthcare workers (CHW) (13). => currently should be placed before demand, to say “to meet the current demand”; or after “are”, to say “are [currently] providing” if the emphasis is that the stated numbers are based on the current state (and of course constantly will shift)

Adding an s to the following terms (if I am reading this right) ... staffing [recruiting/ensuring?] representative[s] of gender, ethnic and economic differences, provision of reproductive and obstetric health services, provision of private breastfeeding space[s] ...

IASC guidelines => writing these abbreviations out at first time

Consistency in using psychosocial or psycho-social

 ... in evidence regarding the effectiveness of inclusion efforts, use of disability and age-disaggregated data over 60 years, cost and benefits of inclusion strategies, and the intersection of disability and older age (8,31). => I assume you may want to include gender in the last part of the sentence, as you are looking at the intersection of gender, disability and older age – otherwise I do not full understand the message of this paragraph, which is also a bit oddly placed between the previous and the subsequent

However, except in the area of SRH services, minimal studies have been [are? Or have been made?] available that have generated evidence on the uptake, practicality, and effectiveness of those interventions.

Given that the protection and gender mainstreaming is a complex phenomenon which involves awareness, actions, behaviour and capacity of a broad spectrum of stakeholders, including policymakers, humanitarian workers and the community, a qualitative approach was chosen to gain an understanding of what interventions in gender and protection mainstreaming are being taken, how practical or effective they are, what is going well and what is not, and how we can further improve the system. => advise to brake into at least 2 sentences

Further, a case study as a qualitative study type is well suited to... => Further, a qualitative case study is well suited to... [may be a shorter way of saying the same thing]

Since no study is available that gives a comprehensive picture of the practicality of gender and protection mainstreaming in health response in the humanitarian context, a case study of the refugee context in Cox’s Bazar was chosen to gain a comprehensive understanding of this complex phenomenon. => sentence could be condensed and avoid repetition

Research philosophy/methods

I am not so familiar with qualitative work, and respective journal requirements. I am therefore not sure if it is common to have subtitles of “research philosophy” and “research setting”, rather methods/approach and setting.

WASH => should be written out the first time it is used, even if commonly understood within the humanitarian sector

 ...it was found that the total sample size of 12 was enough for this study. => sufficient would be a more appropriate word rather than enough, in my experience of journal writing

On the “Data collection instrument” section: In my experience interviews are rather conducted with topics guides, rather than questionnaires; questionnaires tend to be used for quantitative data collection – yet this might be my lack of knowledge within the qualitative methods field

Microsoft Team => Microsoft Teams

Who/how were the interviews transcribed? By teams or by a person? This is not clear. I see this is covered in analysis, I think this can rather fit in process above

“In the case of online interviews, the recording was done by the Microsoft team, and after transcription writing, the recordings were permanently deleted.” => delete “writing” in this sentence

What about translation? As some interviews were conducted in Bangla

Data analysis

This section may need to be re-written/re-ordered, as particularly the first 3 sentences do not flow.

Also the sentences of “Thematic analysis was done for the data collected through in-depth interviews since it is the appropriate method for the analysis of experiences, thoughts and behaviours (36)” is confusing, and I think can be shortened to -> The in-depth interviewed were analysed using thematic analysis, as the methods is appropriate for exploring experiences, thoughts and behaviours – or something along those lines

university-provided P-drive => is not clear to an outsider, maybe you mean to say password-protected server?

Results

Although introduced in the first paragraph “Later, we presented the findings from the in-depth interviews...”, I perceived the switch from report/guideline synthesis based on the reviewed literature to the quotes of from interviews as abrupt. I would advice some (re)introduction when findings from the interviews are presented.

From my experience of reporting on mixed methods (quant and qual), these should either be combined, and then presenting per theme; or into clear sections of e.g. first quant and then qual. I assume the same could be done with the literature insights and the interviews – either they should be integrated and presented together per theme; or separated – or, as said above, some “taking the reader by the hand” is needed when moving to another section.

“It has been found in the health sector quarterly monitoring report that all health facilities (100%, n=136) have a minimum of a female medical professional in the functioning hours of the facility (44).” => this could be formulated with more clarity, who/how/where this has been found, e.g. a report on XX / a review on XX showed that XY. I find “it has been found” vague and would prefer clarity

“Participants expressed that having female staff in the health facility allows women patients/clients to access health care from the facility comfortably; they feel more comfortable speaking and taking part in physical examinations.” => you could cut “from the facility”, and add more elegantly and avoiding repetition “including speaking (up) during and taking part in physical examinations”.

“According to the number of consultations in a medical facility, as several participants mentioned, women use the facilities more frequently than males in the Cox's Bazar context.” => this is confusing, is this based on the number of consultations or based on the reports of several interview participants -> this sentence might need reordering and clarity

Again I would advise to condense “This is a fact for leadership positions as well. Women have less representation in leading positions, like health facility managers or coordinators.” => I would advise something along the lines of: Particularly in leadership positions, such as facility manager or coordinator roles, female staff is scare.

“During the study, “ => do you mean, at the time of the study?

CMR = abbreviation should be clarified at first time use

“Participants also opined that there are only a few training initiatives are undertaken for...” => please check grammar

Even if in a quote, I would advice to use [...] brackets and clarify the meaning of ETAT

PHC = abbreviation should be clarified at first time use

“... stretchers (58% health posts and 86% PHCs), and Crutch...” => delete “and” to make this consistent in your sentence, and make crutches plural and lower case

“...to increase the accessibility of persons with disability and older persons.” => to increase accessibility/ access for persons...

“Although the protection mainstreaming guideline[s?] recommends that latrine design account[s] for children, this is barely taken into account [considered?] during latrine installation.” => please check grammar and avoid repetition of wording if possible, examples provided in breakers as an idea

“... and engineers in 2019 on this.”=> check grammar, it would flow better as “... on this in 2019.”

“A few agencies have some space for children to play or art.”=> create/do art or draw? art is not a verb, please also edit in line 857

Check paragraph of “Targeted interventions toward [for?] vulnerable groups” => some errors in punctuation and spacing, and please consistently use “persons with disabilities” or “people with a disability”

“This can be linked to the fact that 56% of persons...” => this sounds bumpy and confusing; I advise rather something like “this is underlined by / also reported by and assessment conducted by...”

IOM = abbreviation should be clarified at first time use

From my understanding of qualitative work, every quote should have clarity on who stated this, for example with brackets after the quote e.g. (health care specialist), (female expert on GVB) or something alike. Given your field, it might be useful to show the gender of the participant, if allowed and ensuring anonymity

FDMN = abbreviation should be clarified at first time use

Make consistent use of % instead of using the word per cent, unless the first word of a sentence; also numbers above twelve should be written out in numbers, unless the first word of a sentence -> please check this throughout your manuscript

“Frequent staff turnover is another challenge requiring organising training for the newly joined staff.” = please review grammar and flow

“Participants mentioned a good practice that some health facilities are facilitating community health facility support groups with representation from different gender and ages, persons with disability and community leaders. This group owns the health facility and provides feedback through periodic meetings with the facility managers.” => I am not sure I understand this, which group owns the health facility? The community health facility support group that is consulted, or do you mean to say that the group informs the facility? => please clarify

Is there an error in the quote or typo? “Have we considered how we will reach out to those messages who have difficulty in speech and hearing” ? => messengers? Or should the word ; messages’ be deleted, please clarify and it may be possible to use ... in between instead

OPD = abbreviation should be clarified at first time use

“However, experts have opined that this tool cannot be used for children below two years of age.” => is this the main disadvantage and reasons for it not being used? I assume even in high-income countries some disabilities may not be noticed until after the age of 2 years, so I find this comment confusing or the argument may need strengthening or clarity

MMT = should be put into brackets the first time it is written out, to help the reader, please edit

“... include all adults (>18 years)” => I think you want this to say ≥ 18, as otherwise you are not including the 18 year olds, but only those older than 18.

Discussion

I really like the discussion, it flows nicer than the results, and is more concise and clear. It provides good synthesis of the collected data and situates this in the literature.

“Our study found that although it is in principle to engage different layers of the community ...” => please check grammar line 897, I assume you mean “it is a principle to..” but then I would ask by whom, I assume the sector?

“Our study echoes the finding of An After Action Review following a massive fire incident in 2021 that a protection focal person from the protection sector in MMT is a best practice for protection mainstreaming in emergency health response...”=> I do not fully understand what this report found, could this be clarified? What did the focal point do? Can we learn from this situation for others?

Conclusion and Recommendations

“Although the research addressed gender and protection mainstreaming at the local level, many of the findings can be generalised to other similar humanitarian contexts; consequently, the global health and protection clusters..”. => I recommend to break this up and simply start a new sentence with “Consequently, ...”

**Do you want your identity to be public for this peer review?** For information about this choice, including consent withdrawal, please see our Privacy Policy

Reviewer #1: No

Reviewer #2: No

Reviewer #3: No

---

## [Author Response · Author response to Decision Letter 1]

5 Apr 2025

Response to the Academic Editor’s Comments

Comment 1: Ensure that the manuscript meets PLOS ONE’s formatting guidelines.

Response: We have reviewed and updated our manuscript according to PLOS ONE’s style requirements. The file naming convention was aligned with journal standards, and the manuscript was structured according to the specified formatting templates.

Comment 2: Confirm whether the submission contains all raw data required to replicate the study findings.

Response: We confirm that all relevant data supporting our findings are included within the manuscript and its supplementary materials. We have ensured that the values utilised for generating key statistics, graphs, and supporting analyses are accessible.

Comment 3: Ensure that the ethics statement is placed within the Methods section only.

Response: The ethics statement has been relocated to the Methods section, and all duplicate references have been eliminated to adhere to the journal's guidelines.

Response to Reviewer 1’s Comments

Comment 1: Provide more details on the process of obtaining consent and ensuring participant confidentiality and anonymity.

Response: We have expanded the Methods section to clearly explain how informed consent was obtained from participants and the steps taken to maintain confidentiality and anonymity, including secure data storage and anonymized reporting.

Comment 2: Was emotional support available for participants in case of distress?

Response: We have clarified that participants were informed of available psychological support services in case they experienced distress while participating in the study. This information was conveyed to them via the consent process and accompanying letters.

Comment 3: Statistical power considerations should be discussed.

Response: Although our study is qualitative, we have recognised the importance of statistical power in relation to qualitative sample adequacy and have explained the appropriateness of our sample size for achieving thematic saturation.

Comment 4: Consider discussing deviations from normal distribution in the data analysis section.

Response: This qualitative study does not involve variables that necessitate normality testing. However, we have provided further methodological justification for the analytical framework.

Comment 5: The discussion is too descriptive and lacks broader implications.

Response: We have refined the Discussion section to move beyond summarization and incorporate a deeper analysis of implications for gender mainstreaming and protection strategies. New references have been incorporated in each section to enhance the argument.

Comment 6: Ensure that the abstract accurately summarizes the revised manuscript.

Response: The abstract has been revised to ensure consistency with the updated manuscript. It now provides a concise and clear summary of the study’s key contributions.

Response to Reviewer 2’s Comments

Comment 1: The title and study design need better alignment. Why is this considered a case study?

Response: We have clarified the justification for using a case study approach, referencing key methodological sources. Given the specificity of the refugee camp setting, our study follows a case study methodology rather than a general qualitative approach. Methods section has been updated with this clarification.

Comment 2: The term “Gender and Protection Mainstreaming” may be unfamiliar; consider alternative wording.

Response: We have included a formal definition from sectoral and global cluster guidelines to ensure clarity for a broader audience. We have preserved the original terminology to ensure consistency with international humanitarian frameworks.

Comment 3: The methods section lacks details about data analysis and potential bias.

Response: We have expanded the Methods section to specify participant selection, data analysis techniques, and measures taken to reduce bias, including reflexivity practices and triangulation techniques. A limitation related to bias has been incorporated into the Limitations section.

Comment 4: The Results section needs better participant details and demographic information.

Response: We have provided additional details about participant demographics, roles, and rationale for selection in the results section.

Comment 5: The conclusion should be reframed to better reflect the study’s findings without overgeneralization.

Response: Conclusion has been revised to reflect the study's contributions, avoiding overstatements and emphasising the relevance of findings to humanitarian contexts.

Comment 6: The manuscript lacks clear definitions and structure in the introduction and background.

Response: The Introduction has been revised to clearly define key concepts, including gender and protection mainstreaming. Coherence between the background and objectives sections has been enhanced.

Response to Reviewer 3’s Comments

Comment 1: Improve sentence structure and clarity in various sections, particularly the Results.

Response: We have conducted a thorough language review and proofreading process to ensure clarity and conciseness throughout the manuscript. Sentences have been shortened for improved readability, and redundancies have been eliminated.

Comment 2: Define all abbreviations at their first use.

Response: Abbreviations such as IASC, WASH, PHC, OPD, MMT, FDMN have been written out in full upon their first mention.

Comment 3: The transition between the literature synthesis and interview findings is abrupt.

Response: We have improved transitions between sections, ensuring that the shift from literature findings to qualitative insights is smoother and more intuitive.

Comment 4: Improve clarity in some quoted excerpts.

Response: We have revised quotes for clarity while maintaining their original meaning. Bracketed clarifications have been included where necessary to enhance comprehension.

Comment 5: Ensure consistency in numerical formatting and terminology.

Response: We have standardized numerical representations (e.g., using % instead of "percent") and ensured consistent terminology use throughout the manuscript.

Comment 6: Improve grammar and wording in specific sentences.

Response: All suggested grammatical revisions and rewordings have been implemented. This involves revising unclear sentences to enhance readability.

Note: We have added the qualitative dataset (thematic analysis) - as supplementary information. We removed sensitive information (e.g. agency name) that could identify the participant.

---

## [Decision Letter · Decision Letter 1]

The Practice of Gender and Protection Mainstreaming in Health Response in Humanitarian Crisis - A Case Study from the Refugee Camps in Cox’s Bazar, Bangladesh

PONE-D-24-39518R1

Dear Charls Erik Halder,

We’re pleased to inform you that your manuscript has been judged scientifically suitable for publication and will be formally accepted for publication once it meets all outstanding technical requirements.

Kind regards,

Md. Feroz Kabir, BPT, MPT, MPH, BPED, MPED

Academic Editor

PLOS ONE

Additional Editor Comments (optional):

Reviewers' comments:

Reviewer's Responses to Questions

**Comments to the Author**

Reviewer #4: All comments have been addressed

Reviewer #5: All comments have been addressed

2. Is the manuscript technically sound, and do the data support the conclusions?

Reviewer #4: Yes

Reviewer #5: Yes

3. Has the statistical analysis been performed appropriately and rigorously?

Reviewer #4: Yes

Reviewer #5: I Don't Know

4. Have the authors made all data underlying the findings in their manuscript fully available?

Reviewer #4: Yes

Reviewer #5: Yes

5. Is the manuscript presented in an intelligible fashion and written in standard English?

Reviewer #4: Yes

Reviewer #5: Yes

Reviewer #4: Thanks for your point-by-point response. Now it looks well and makes sound. Still, the English grammar should be revised all through the paper.

Reviewer #5: Authors have responded the correctly to the comments raised by the reviewers and overall quality of the manuscript has been improved since then. This paper is now in a acceptable form. Thank you the authors for their hard work.

**Do you want your identity to be public for this peer review?** For information about this choice, including consent withdrawal, please see our Privacy Policy

Reviewer #4: **Yes: ** Mohammad Mohinul Islam

Reviewer #5: No

---

## [Editor Report · Acceptance letter]

PONE-D-24-39518R1

PLOS ONE

Dear Dr. Halder,

I'm pleased to inform you that your manuscript has been deemed suitable for publication in PLOS ONE. Congratulations! Your manuscript is now being handed over to our production team.

Kind regards,

on behalf of

Dr. Md. Feroz Kabir

Academic Editor

PLOS ONE